# A Perfect Plastic Material for Studies on Self-Propelled Motion on the Water Surface

**DOI:** 10.3390/molecules26113116

**Published:** 2021-05-23

**Authors:** Richard J. G. Löffler, Martin M. Hanczyc, Jerzy Gorecki

**Affiliations:** 1Institute of Physical Chemistry, Polish Academy of Sciences, Kasprzaka 44/52, 01-224 Warsaw, Poland; rloffler@ichf.edu.pl; 2Laboratory for Artificial Biology, Centre for Integrative Biology (CIBIO), University of Trento, Polo Scientifico e Tecnologico Fabio Ferrari, Polo B, Via Sommarive 9, 38123 Povo, TN, Italy; martin.hanczyc@unitn.it; 3Chemical and Biological Engineering, University of New Mexico, Albuquerque, NM 87131, USA

**Keywords:** camphene, camphor, polypropylene, surface tension, self-propelled motion

## Abstract

We describe a novel plastic material composed of camphene, camphor, and polypropylene that seems perfectly suited for studies on self-propelled objects on the water surface. Self-motion is one of the attributes of life, and chemically propelled objects show numerous similarities with animated motion. One of important questions is the relationship between the object shape and its motility. In our paper published in 2019, we presented a novel hybrid material, obtained from the solution of camphor in camphene, that allowed making objects of various shapes. This hybrid material has wax-like mechanical properties, but it has a very high tackiness. Here, we report that a small amount of polypropylene removed this undesirable feature. We investigated the properties of camphor–camphene–polypropylene plastic by performing the statistical analysis of a pill trajectory inside a Petri dish and compared them with those of camphor-camphene wax. The plastic showed the stable character of motion for over an hour-long experiment. The surface activity of objects made of plastic did not significantly depend on the weight ratios of the compounds. Such a significant increase in usefulness came from the polypropylene, which controlled the dissipation of camphor and camphene molecules.

## 1. Introduction

Studies on the time evolution of objects, self-propelled on aqueous surfaces, attract significant scientific attention as an interesting example of systems in which the transformation of chemical energy to kinetic energy creates complex and even life-like behavior [1]. The motion of a camphor piece on the water surface is a classic and simple example of such energy conversions [2,3,4]. Under typical experimental conditions, camphor quickly dissipates as a result of sublimation, dissolution, and the formation of a surface layer. Camphor is a crystalline terpenoid that poorly dissolves in water (1.2 g·dm${}^{-3}$ [5]). Thus, the dissolution of camphor into the water volume does not have a significant influence on the self-motion. However, camphor is amphiphilic, and the water surface tension is a decreasing function of the camphor surface concentration. The key element enabling the self-motion of a camphor piece is the dynamics of the camphor layer at the air–water interface [6,7,8]. At room temperature, the layer is unstable because camphor molecules are continuously evaporating from the layer to the air [6]. Let us consider a camphor piece lying on the water surface. There is a continuous flux of camphor molecules that modifies the surface tension from the source to the air through the surface layer. The surface tension remains low in the region close to the camphor piece, where the concentration of camphor is high. The surface tension has been estimated as 65 mN/m proximal to the source and 70 mN/m distal from the camphor [9,10,11].

The camphor evaporation from the surface is stochastic, and areas characterized by a low surface concentration (and thus with a high surface tension) can appear close to the camphor source. The force acting on the camphor piece is directed toward these areas and can initiate self-propelled motion [7]. If a camphor piece moves from its original position, then initially, it shifts to the area characterized by large gradients of the camphor surface concentration. This significantly increases the speed of the piece. The physical mechanism described above is general, and it also explains the behavior of objects made of other substances exhibiting similar surface activity.

The explanation of a relationship between the symmetry and the geometrical shape of a solid, self-propelled object and the character of its motion has stimulated abundant scientific activity [12,13,14,15,16,17,18]. It can be expected that shape and motion are correlated. Obviously, the object shape defines the resistance to its motion on the surface. On the other hand, the source also determines the local outflow of surface active molecules from the source and the force acting on the object because it can be calculated as the interval of the surface tension over the object contour [8,19].

Experimental studies on the complexity of self-propelled motion of solid objects characterized by different shapes are important because they are a sensitive tool for the verification of the applicability and quality of theoretical models for self-propelled motion. However, it is not easy to make camphor pieces of the required shapes, especially when many objects with the same shape are needed for an experiment. A few techniques have been used to transform the camphor grains into objects of a defined shape. The most popular is pressing camphor granules together in a pill maker, but this technique mainly applies to small camphor disks [20]. An inert porous support material, like a paper membrane [21] or agar gel [22,23], can be impregnated with a camphor solution, but this technique is not ideal because the process is hard to control and may produce a non-homogeneous distribution of camphor. Moreover, the saturated objects contain smaller amounts of camphor than pills, limiting surface activity to the order of minutes [16].

In our recent paper [24], we described a new easily shapable waxy material, obtained from a hot (150 ${}^{{}^{\circ}}$C) homogeneous mixture of camphor and camphene in liquid form, cooled to room temperature. Camphene is a wax-like sticky terpenoid with low solubility in water (0.004 g·dm${}^{-3}$ [25]) that exhibits similar, but decreased [24,26], surface activity compared to camphor. Just like camphor, it forms an unstable, evaporating layer that decreases the surface tension depending on the surface concentration of corresponding molecules. A typical pill made of the camphene–camphor mixture (camphene–camphor wax) exhibits surface activity for over 1 h while the object itself decreases in size. Thus, the duration of its self-motion is similar to that of a pill made of pure camphor. We observed that the character of motion can be controlled by the camphor-to-camphene weight ratio. The camphene–camphor wax material seems perfectly suited for studies on the self-propelled motion of different shapes over a long time. It has, however, one undesirable feature: the material is very sticky. We found that there are just a few substrates (nitrile, silicone, or latex) to which it does not stick, and these materials have to be used to manipulate objects made of the camphene–camphor wax. Additionally, at a higher ratio of camphene, the material is too soft to form it into precise shapes, using a pill press for example.

In this paper, we present an important improvement of the camphene–camphor wax. We discovered that a small amount of polypropylene added to the camphene–camphor mixture reduces the material tackiness, making the material much easier to work with. The ternary camphene–camphor–polypropylene hybrid material (camphene–camphor–polypropylene plastic) can be easily processed and shaped on most clean substrates. Here, we describe how to make the mixture and discuss the character of the self-propelled motion of pills made of the new ternary material. We compared them to pills of the same geometry made of the previously studied camphene–camphor wax. We demonstrated the usefulness of camphene–camphor–polypropylene plastic in experiments with the self-motion of single or multiple interacting self-propelled objects of a non-trivial shape. The time evolution observed for a system of rods that form clusters in which end-to-end attractions are dominant seems to be an interesting system for further studies.

## 2. Materials and Methods

Samples at different weight ratios of compounds were prepared by weighing off commercially available (1R)-(+)-Camphor (98% purity, CAS: 464-49-3, Sigma-Aldrich, Darmstadt, Germany), camphene (95% purity, CAS: 79-92-5, Sigma-Aldrich, Darmstadt, Germany), and polypropylene in the form of pellets (CAS:9003-07-0, Sigma-Aldrich, Product Number 427861) into a 50 mL beaker containing a magnetic stir bar and covered with a larger beaker in order to prevent excessive evaporation during heating. The mixture was then placed on a hot plate set to 250 ${}^{{}^{\circ}}$C while stirring until all the polypropylene was dissolved in the liquid camphene–camphor mixture (between 20 and 40 min, depending on the ratio). As the material is sticky shortly after solidification, it was poured into a vessel lined with nitrile, latex, or silicone sheets. Pills of this material can be pressed out using a pill maker. Flat pieces of the camphene–camphor–polypropylene plastic can be formed and cut to the required shape. Rods made of this material can be obtained by pressing out the camphene–camphor–polypropylene plastic, while it is hot and soft at ca. 60 ${}^{{}^{\circ}}$C, from a syringe through a hole with the selected diameter. This method was used to obtain a long camphene–camphor–polypropylene plastic filament cut into shorter rods of the lengths required for the experiments (See Section 3.4).

Experiments demonstrating the properties of the camphene–camphor–polypropylene plastic were performed at 23 ± 1 ${}^{{}^{\circ}}$C in a Petri dish with a diameter of 12 cm for material with a 10% weight ratio of polypropylene, containing 50 mL of water purified using a Millipore ELIX5 system. Therefore, the water level was 0.44 cm. For the material with the 5% weight ratio of polypropylene, a Petri dish of 11 cm in diameter was used. In this case, the water level was 0.53 cm. The Petri dish was illuminated from below while recording from above using a mounted digital camera (NEX VG20EH, SONY (25 fps, time between frames Δt=0.04 s) or Logitech C920 Webcam (30 fps, time between frames Δt=1/30 s)). The studied self-propelled objects were marked with black stickers that increased the contrast and helped to trace the trajectory. The movies of the performed experiments were separated into frames using the ffmpeg program [27]. The locations and orientations of objects on individual frames were extracted using ImageJ [28] or with Mathematica [29]. The information on the object time evolution (trajectory and speed as functions of time, statistical information on object locations, and the distribution of the speed) was analyzed with Mathematica.

The movie was a sequence of *M* frames, and the m-th frame corresponded to the time $t_{m}=m*\Delta t$ (0≤m<M). The basic information extracted from the movie was the positions of the marker center $(x_{m}$, $y_{m})$. This information allowed us to investigate the object’s trajectory and velocity. A similar analysis for pills made of camphene–camphor wax was reported in [24]. This method allowed for the comparison of the properties of the new camphene–camphor–polypropylene plastic with the properties of the camphene–camphor wax previously described. For both types of materials, we investigated the shape of the trajectory, the distribution of pill locations on the Petri dish, and both the pill speed as a function of time and the distribution of the speed values.

The system was characterized by a circular symmetry. Therefore, the statistics of pill positions on the water surface can be characterized by the distribution function of pill centers g(r) that measures the difference between the observed distributions of pill locations with respect to the dish center and the uniform distribution, with the dish radius as *R* and the pill diameter *d*. In the case of uniformly distributed pill positions, the number of frames in which the pill center was observed within the fragment *S* of the water surface $n_{0}\left(S\right)$ is:(1)$$n_{0}\left(S\right)=M\cdot \frac{∥S∥}{\pi {\left(R-d/2\right)}^{2}}$$
where ∥S∥ is the area of the fragment S. For example, if we considered a ring around the dish center U[r,r+Δr] with the inner radius *r* and the outer radius r+Δr, then the number of frames at which the pill center is located between the two radii is described by:(2)$$n_{0}\left(U\left[r,r+\Delta r\right]\right)∼\frac{2Mr\Delta r}{{\left(R-d/2\right)}^{2}}$$

However, the real distribution of pill locations can be different from the uniform one. Let n(U[r,r+Δr]) denote the number of frames of the movie in which the pill center is located within the ring. The function g(r) is defined as:(3)$$g\left(r\right)≅\frac{n(U[r,r+\Delta r])}{n_{0}\left(U\left[r,r+\Delta r\right]\right)}$$
where Δr is reasonably small.

In our study on camphene–camphor wax [24], we demonstrated that g(r) depends on the camphene–camphor weight ratio. For the pure camphene, g(r)=0, if |R−r|<10 mm, meaning that the pill never came closer than 10 mm from the edge. The probability of finding a pill close to the dish edge increased with the amount of camphor in the mixture. For equal weights of camphene and camphor, the pill positions were uniformly distributed on the whole surface. Here, we also used the function g(r) for the quantitative characterization of the character of motion of pills made of camphene–camphor–polypropylene plastic for different weight ratios of the compounds.

Having the positions of the pill center at successive frames of the movie ($x_{m}$, $y_{m}$), we can calculate its speed at time $t_{m}$ as:(4)$$v\left(t_{m}\right)=\frac{\sqrt{{\left(x_{\left(m+1\right)}-x_{\left(m-1\right)}\right)}^{2}+y_{\left(m+1\right)}-y_{\left(m-1\right)}{)}^{2}}}{2\Delta t}$$

In the previously reported experiments with camphene–camphor wax, we observed a qualitative difference between P(v) for pills made of pure camphene and P(v) for pills made of a camphene–camphor wax [24]. In the first case, the distribution was Poissonian; in the second, it was bimodal. The magnitude of the peak corresponding to a high velocity was growing with the weight ratio of camphor. Therefore, we studied the pill speed as a function of time and the probability distribution of the speed values P(v).

## 3. Results and Discussion

The camphene–polypropylene and the camphene–camphor–polypropylene plastics were sticky, waxy solids after they were cooled down to ambient temperature. The hardness of the plastics increased with the amount of camphor and/or polypropylene. The stickiness mentioned in [24] decreased with the amount of dissolved polypropylene.

### 3.1. Time Evolution of Pills Made of Camphene–Polypropylene Plastics

First, we tested pills made of camphene–polypropylene plastics. We can expect [24] that the self-propelled motion of such pills qualitatively differs from that observed for camphene–camphor–polypropylene plastic.

We observed that a pill, just after placement onto the water surface, moved at high speeds and slowed down to approach a stable speed. Such an effect is illustrated in Figure 1a,b for a 4 mm pill made of 90% by weight of camphene and 10% by weight of polypropylene moving inside a Petri dish with a 12 cm diameter. The trajectory in Figure 1a shows the first 60 s of motion after the pill was placed on the water surface. The yellow disk marks the initial position of the pill, and the green one shows its location after 60 s. The corresponding pill speed as a function of time is presented in Figure 1b. It reached a stable value around t=40 s. Figure 1c shows a typical trajectory of a camphene–polypropylene pill observed over a longer time. Here, the blue line illustrates the trajectory observed in the time interval [66 min, 70 min], while the red line corresponds to the time interval [70 min, 79 min]. The yellow, green, and magenta disks mark the pill positions at t=66,70, and 79 min. The length of the blue line, representing 4 min of motion, is much longer than the red one covering 9 min. This indicates that the speed of pills made of camphene–polypropylene plastic can significantly decrease after 1 h of activity.

For the following analysis of camphene–polypropylene pill motion, we restricted motion analysis to the time interval [1 to 61 min].

Figure 2, Figure 3, Figure 4 and Figure 5 summarize three independent experiments in which the motion of a 4 mm pill made of 95% by weight of camphene and 5% by weight of polypropylene was observed. The dish diameter was 11 cm. The trajectories and the distances between the pill center and the dish center are shown in Figure 2. The results were obtained by the analysis of individual frames recorded at a rate of 30 fps. In each experiment, we observed transitions between the rotational motion along the edge of the dish and the random motion inside the dish. Neither of these types of motion seemed to dominate the evolution. In one case (the red trajectory), the random motion covering the dish interior appeared after almost a 40 min-long stable rotation. In another case (the blue trajectory), stable, elliptic-shaped motion crossing the dish center switched to rotation after almost 50 min of evolution. In the third case (the green trajectory), the motion was strongly dominated by rotations. The dominating character of rotation can be seen in Figure 3 as a strong peak of the radial distribution function g(r) around 3.7 cm from the dish center. A similar radius of stable rotation was observed in all experiments. As reported in [24], a pill self-propelled by camphene only, does not come closer to the dish edge than 1.3 cm. Such a character of motion was also observed in experiments with camphene pieces [26]. The stable motion along the edge can be explained as follows: We can assume that the surface active molecules are symmetrically dissipated around the disk. Those that move towards the dish center are dispersed over a larger area than those that move towards the dish edge because the edge restricts their motion. As a result, one can expect a higher concentration of surface active molecules in the region between the pill and the dish edge than in the central part of the dish. The gradient of concentration translates into the gradient of surface tension that repels a pill away from the edge.

Figure 4 presents the speed of a pill as a function of time for the trajectories illustrated in Figure 2. The majority of the measured speed values were below 1.5 cm/s, but there were occasional high-speed bursts approaching 3 cm/s. Such increases in speed did not strongly correlate with the character of motion. Even so, the speed of characterizing the stable rotation was, on average, a bit lower than that corresponding to the random motion. The speeds averaged over the observation time were 0.46 cm/s, 0.47 cm/s, and 0.21 cm/s for the “red”, “blue”, and “green” experiments, respectively. These numbers were slightly lower than the average speed measured in the experiments with pills made of pure camphene (0.6 cm/s in [24]).

Figure 5 shows the probability distribution of the speeds for a pill made of 95% camphene and 5% polypropylene. The main figure shows the results of individual experiments. The distribution of the speed values observed in all experiments collectively is shown in the insert. It had a Poissonian character, which confirmed the observations for self-propelled objects made of pure camphene [24,26].

To study the influence of the polypropylene content on the self-motion of a camphene pill, we performed three independent experiments with a 4 mm pill made of plastic composed of 90% by weight of camphene and 10% by weight of polypropylene. The dish diameter was 12 cm. The trajectories and the distances between the pill center and the dish center are shown in Figure 6. The results were obtained by the analysis of individual frames recorded at a rate of 25 fps. The character of motion was similar to that observed for pills made of 95% camphene and 5% polypropylene. In each experiment, we observed random motion inside the dish. Moreover, in two of three experiments, we also observed a stable rotation along the dish edge. The dominating character of rotation can be seen in Figure 7 as a strong peak of the radial distribution function g(r) at a 4.0 cm distance from the dish center. The increase in the peak position if compared with g(r) shown in Figure 3 can be attributed to a larger diameter of the dish.

Figure 8 presents the speed of a pill as a function of time for the trajectories illustrated in Figure 6. Here, all measured speed values were below 1.5 cm/s, and we did not observe any spikes. The speeds averaged over the observation time were 0.38 cm/s, 0.79 cm/s, and 0.54 cm/s for the “red”, “blue”, and “green” experiments, respectively. These numbers were in good agreement with the average speed measured in experiments with pills made of pure camphene (0.6 cm/s in [24]). Figure 9 shows the probability distribution of the speeds for a pill made of 90% camphene and 10% polypropylene. The large figure shows the results of individual experiments and, in the insert, the distribution of all speed values observed in the three experiments. The probability distributions of speed observed in the “red” and “green” experiments had a Poissonian form. The third distribution was Gaussian, with a maximum of around 0.8 cm/s. As a result, the distribution of all values of speed was almost flat for speeds <1 cm/s.

### 3.2. Self-Motion of a Pill Made of Camphene–Camphor–Polypropylene Plastic

The results presented above suggested that the self-motion properties did not depend on the amount of polypropylene in the pill. Next, we considered if and how the weight ratio between camphene and camphor modified the character of motion. We observed that for camphene–camphor–polypropylene plastic, the stabilization of pill speed required a much longer time than for camphene–polypropylene plastic (cf. Figure 1b and Appendix A included in the Appendix A). Therefore, in the following analysis, we considered an hour-long motion, starting five minutes after a pill was placed on the water surface.

As an example of the motion of pills made of camphene–camphor–polypropylene plastic, we show the results for 45%, 45%, and 10% weight ratios between the components (Figure 10, Figure 11, Figure 12 and Figure 13). In the experiment, the dish diameter was 12 cm, and the trajectory was recorded at a rate of 25 fps. The trajectories and the distances between the pill center and the dish center obtained in the three independent experiments are shown in Figure 10. The red trajectory covers the whole dish. The time evolution of the distance between the pill center and the dish center indicates that, at the beginning, the pill rotated along the dish edge with the dispersion of radii exceeding 1 cm (also observed in the “green” experiment). Next, the dispersion of distance increased, and after 2000 s, the pill started to move irregularly on the whole surface. In the second and the third experiments (blue and green curves), we observed the rotational motion of the pill along the dish edge. In the “blue” experiment, the radius of the motion was in a narrow range (between [4.9 cm and 5.1 cm]) and remained stable. In the “green” experiment, the initial (circa 2 min long) stable rotation with a narrow dispersion of radii changed into a complex rotation characterized by a large dispersion of radii that increased over time. Results in Figure 10 show that the pill came much closer to the wall than in the experiments with camphene–polypropylene plastic. Nevertheless, we observed no contact between the pill and the wall. The largest recorded distances between the pill center and the dish center were 5.62, 5.23, and 5.41 cm, respectively; thus, the distance between the pill edge and the dish wall was always larger than 2 mm. Figure 11 shows the radial distribution function g(r) calculated from the distances measured in separate experiments. The results of individual experiments were strongly dominated by the type of observed motion.

Figure 12 presents the speed of a pill made of 10% polypropylene, 45% camphene, and 45% camphor as a function of time. The average values of the speed observed for a pill made of 10% polypropylene, 45% camphene, and 45% camphor were 5.28, 7.63, and 7.57 cm/s for the red, blue, and green function, respectively. In all experiments, we observed an initial decrease of speed, but it did not exceed 25% of the stationary value. As expected, speed was the most stable for the most uniform trajectory (rotation along the dish wall). Figure 13 shows the probability distribution of speeds. The large figure shows the results of individual experiments. The distribution of speed values observed in all experiments is shown in the insert.

A strong influence of the proportions between amounts of camphor and camphene on the speed of a pill made of camphene–camphor wax was observed [24]. We did not observe such an effect for camphene–camphor–polypropylene plastics. The properties of pill self-motion did not depend on the camphene/camphor weight ratio in the range from 5/1 to 1/1. Therefore, here in the main text of the paper, we presented just a single analysis of the time evolution (Figure 10, Figure 11, Figure 12 and Figure 13). The results of experiments with other camphene/camphor ratios are given in the Appendix A.

In all cases, we observed the rotation along the dish wall with a narrow dispersion of radii $\sigma_{r}<0.5$ cm, a complex rotation mode with a large rotation of radii $\sigma_{r}∼3$ cm, and a complex motion on the whole dish surface as metastable modes of pill motion. Contrasted with the results of camphene–camphor waxes, there was no indication that a pill made of camphene–camphor–polypropylene plastics came closer to the wall when the camphor concentration increased. In all experiments, except a single one (the red trajectory in Figure 10), the pill center was more than 6 mm away from the dish edge. No systematic decrease in the minimum approach distance with increasing camphor concentration was observed.

The probability distributions of the speed observed in a single experiment (cf. Appendix A) were Gaussian in the majority of cases. There were also experiments in which we recorded a bimodal speed distribution (cf. Appendix A). The bimodal distribution of the speed was also observed for camphene–camphor waxes and camphor weight ratios of 33% and 37% [24]. For the waxes, the low-speed and the high-speed maxima of the probability distribution appeared at 2 cm/s and 12 cm/s, respectively. For camphene–camphor–polypropylene plastic, the difference between the locations of maxima was much smaller and did not exceed 3 cm/s. In general, the speeds observed for camphene–camphor–polypropylene plastics were smaller than for camphene–camphor waxes. For the waxes, speeds exceeding 10 cm/s were observed with a high probability. On the other hand, the speed of a pill made of camphene–camphor–polypropylene plastics hardly exceeded 10 cm/s. The decrease in speed in comparison to camphene–camphor waxes can be explained by the dissipation of camphene and camphor from the pill surface within the neglected initial 300 s of the experiment. For longer times, the passivating polypropylene layer modulated the outflow of surface active molecules. The results presented in Figure 10, Figure 11, Figure 12 and Figure 13 and in the Appendix A indicate that the self-propelled motion of a pill made of camphene–camphor–polypropylene plastics was independent of the camphene/camphor/polypropylene weight ratio.

### 3.3. Long-Term Behavior of a Pill Made of 10% Polypropylene, 45% Camphene, and 45% Camphor

In this section, we discuss the long-time motion of a 4 mm pill made of camphene–camphor–polypropylene plastic with a 45% camphene, 45% camphor, and 10% polypropylene weight ratio on the water surface. The motion was observed in a Petri dish with 12 cm diameter and for a 0.44 cm water level. Figure 14a presents the distance between the pill center and the dish center as a function of time for an experiment recorded for 5.5 h. The colors of the curve mark different types of motion illustrated in Figure 14b–d. In the red part (Figure 14b, t∈[3200 s, 3700 s]), the pill randomly rotated on the whole dish surface. Following that, the type of motion changed to a stable rotation along the dish edge with a ∼7 s period. The transition from an irregular motion to a stable rotation was predicted in theoretical studies [30]. As seen in Figure 14a, the radius of rotational motion slightly increased with time due to the reduced outflow of surface active molecules. The trajectories of pill motion in long-time intervals t∈[4400 s, 6000 s] (Figure 14c) and t∈[16,000 s, 20,200 s] (Figure 14d) showed that the rotation was very stable. Figure 14d includes over 500 individual cycles, and we did not observe any significant dispersion of radii.

The speed of a pill as a function of time is plotted in Figure 15. The total distance traveled by the pill exceeded 1 km (!), which is probably a record in experiments with self-propelled objects. In the rotational mode, the speed decreased slowly (from 7 cm/s to 4 cm/s during a 4 h period) as a result of the reduced dissipation of camphor and camphene. However, we hardly noticed any decrease during the first hour of the rotational motion. We believe that such a high stability of motion makes camphene–camphor–polypropylene plastic an ideal candidate for experiments with self-propelled objects.

### 3.4. Motion of Self-Propelled Rods

In the previous section, we discussed the self-propelled motion of a pill made of camphene–camphor–polypropylene plastic. However, pill-shaped objects can be made of camphene or camphor in a pill maker. The advantage of new materials like camphene–camphor wax or camphene–camphor–polypropylene plastic lies in the fact that they can be formed into non-trivial shapes. Here, we show the results for the self-propelled motion of rods made of plastic composed of a 10% polypropylene, 45% camphene, and 45% camphor weight ratio. Figure 16 shows types of motion observed in a single experiment with a rod (10 mm long and 2 mm diameter) moving on the water surface in a Petri dish with a 5 cm diameter. Figure 16a–d illustrates how the type of motion changes with time. Each subfigure illustrates the positions of the rod center (the red curve) and the rod orientation (color lines) during a 17 s-long evolution. For each subfigure, the time is indicated as the line color. The color coding is defined on the bar below the figures. After the rod was placed on the water surface, the fast rotation dominated. In this mode, the center of the rod hardly moved (cf. Figure 16a, recorded 72 s after the rod was placed on the surface). A few minutes later, the character of the motion changed. The rod showed irregular motion in the whole dish (Figure 16b, 240 s after the experiment initiation). Such a mode was stable for a few minutes and transformed into rotation along the dish edge. In this mode, the angle between the rod orientation and the vector linking the dish center with the rode center did not change significantly, so the rode rotation was synchronized with rotation along the edge. Initially, the trajectory of the rod center had a polygonal shape (Figure 16c, 920 s after the experiment initiation); for longer times, it became circular (Figure 16c, 1200 s after the experiment initiation). The explanation of such a sequence of modes is an interesting challenge for theoretical studies we plan to undertake.

The fact that rods can be easily made with camphene–camphor–polypropylene plastic encouraged us to study the time evolution of systems containing multiple rods. A few snapshots illustrating the evolution of 20 rods that were 1 cm long on the water surface in a Petri dish with a diameter of 12 cm are shown in Figure 17. The rods were made of 10% polypropylene, 45% camphene, and 45% camphor. The movie showing the first 4 min of the time evolution is included in the Appendix A as 20-rod-evolution.avi. In the beginning, the rods were moving individually on the surface, followed by the formation of clusters, increasing in size as time progressed. It can be seen that the dominant form of rod attraction was the one between the rod ends. We also observed attraction between the end of one rod and the center of the long edge of another (cf. Figure 17c,d). This type of attraction seemed weaker than the end-to-end one because the pairs of rods attached end-to-center were less frequently seen than those sticking at the ends. The geometry of interaction makes sense if one considers that the deposition of surface active material from the rods was asymmetrical such that more material was deposited at the long edges than at the ends. Thus, the repulsion between rods was larger at the long edge. The clusters were dynamically changing in time. They could merge or separate into fragments. Figure 18 illustrates the number of rods grouped into clusters of a given size. It can be noticed that the numbers of separated rods and small clusters decreased with time, and most rods were trapped in a small number of large clusters.

## 4. Conclusions

The motile behavior of self-propelled objects on liquid surfaces is an interesting manifestation of the transformation of chemical into kinetic energy. Problems such as the relationship between the shape of a self-propelled object and the character of its motion have attracted significant attention from the academic community. However, when using common materials, like camphor, it is difficult to make objects of non-trivial shapes. An easy-to-shape camphene–camphor wax was described in our previous study [24]. However, the material was very sticky, and its properties were strongly dependent on the component ratio. By adding polypropylene to camphene–camphor wax, we successfully eliminated some of the drawbacks that made the wax difficult to work with. In this paper, we described the properties of a new hybrid material: camphene–camphor–polypropylene plastic. The material seems to be perfect for experiments on self-propelled motion. The camphene–camphor–polypropylene plastic can be easily shaped and has a low tackiness. The material does not lose its surface activity after melting, so molds or potentially even 3D printers can be used to form self-moving objects of highly controlled and diverse shapes. Objects made of plastic showed unprecedented, hours-long stability of self-propelled motion. The self-propelled motion of pills made with the plastic did not significantly depended on the composition of components, so potential mistakes in the plastic preparation should not quantitatively affect the observed motion.

We think that the key to our success was the presence of polypropylene in the plastic. The polypropylene moderated the migration of camphene and camphor molecules from the interior of an object towards its surface. A short time after the plastic was made, its surface was free of camphene molecules because they evaporated. Therefore, the concentration of the sticking component on the plastic surface was small enough to reduce the tackiness to a reasonable level. During the self-propelled motion, the dissipation of surface active molecules was also modulated by the polypropylene. For the reported polypropylene ratio, this dissipation was high enough to ensure self-propelled motion on the water surface, but on the other hand, it was low enough to support sustained motion over long periods of time.

## Figures and Tables

**Figure 1 molecules-26-03116-f001:**
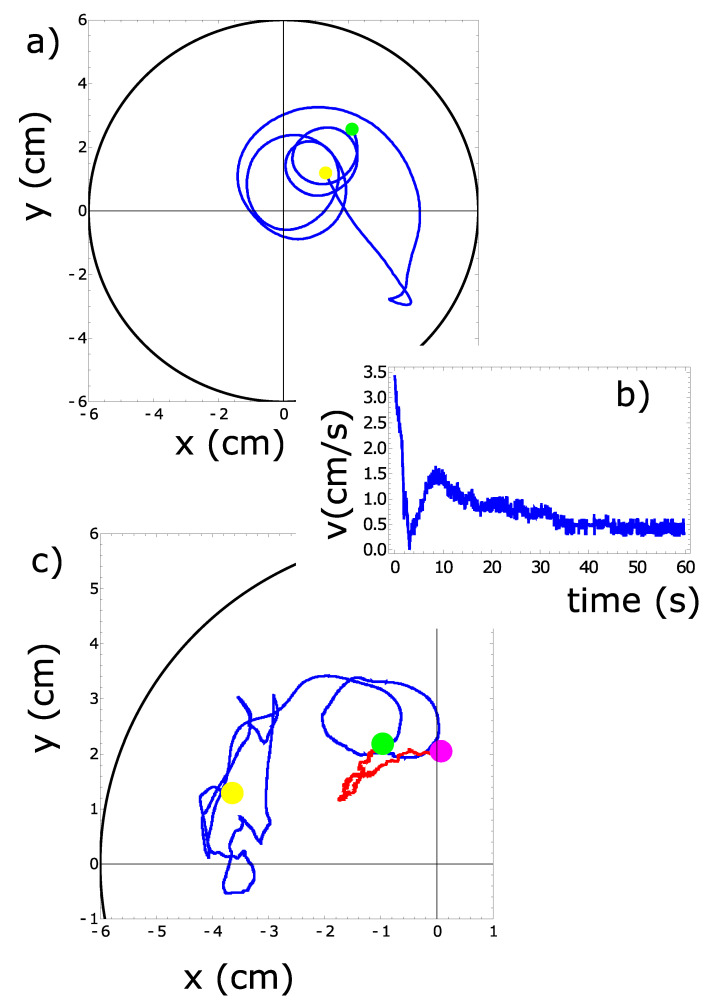
The motion of a 4 mm pill made of 90% by weight of camphene and 10% by weight of polypropylene observed at short and long times. (**a**) The trajectory of the first 60 s of motion after the pill was located on the water surface. The yellow disk marks the initial position of the pill, and the green one shows its location after 60 s. (**b**) The pill speed measured for the trajectory shown in (**a**). (**c**) The blue and red lines show the trajectory observed in the time intervals [66 min, 70 min] and [70 min, 79 min] after the pill was placed on the water surface, respectively. The yellow, green, and magenta disks mark the pill positions at t=66,70, and 79 min.

**Figure 2 molecules-26-03116-f002:**
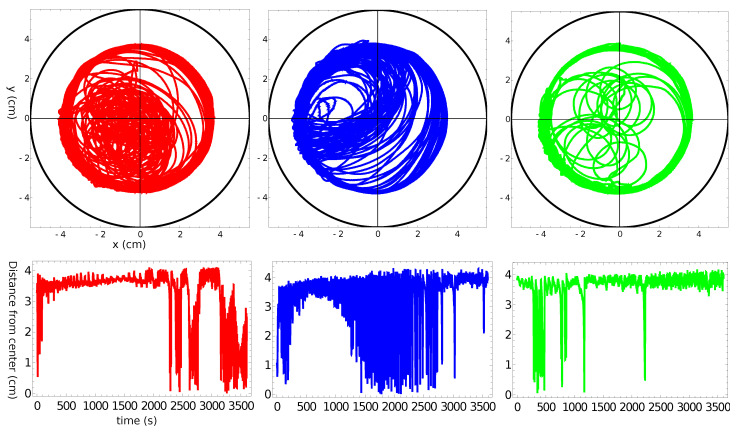
The upper three figures show the trajectories of a 4 mm pill made of 95% camphene and 5% polypropylene in a Petri dish with an 11 cm diameter. The black circle indicates the location of the dish wall. The distance between the pill center and the dish center as a function of time is plotted in the figures located below the corresponding trajectories.

**Figure 3 molecules-26-03116-f003:**
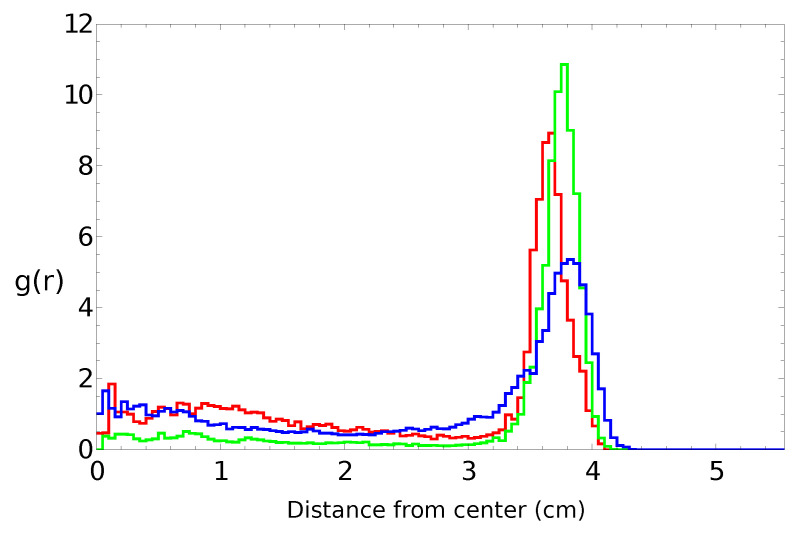
The radial distribution function g(r) of the positions of pill centers calculated from the distances illustrated in Figure 2. The same color is used to draw the radial distribution function and the trajectory.

**Figure 4 molecules-26-03116-f004:**
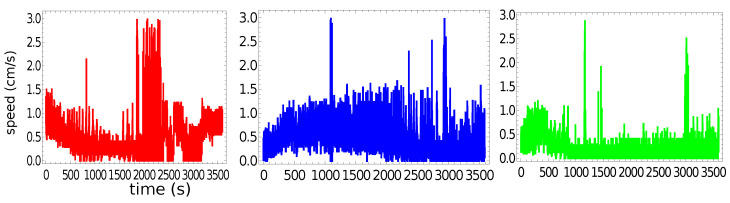
The speed of a pill made of 95% camphene and 5% polypropylene as a function of time, as observed in the three independent experiments.

**Figure 5 molecules-26-03116-f005:**
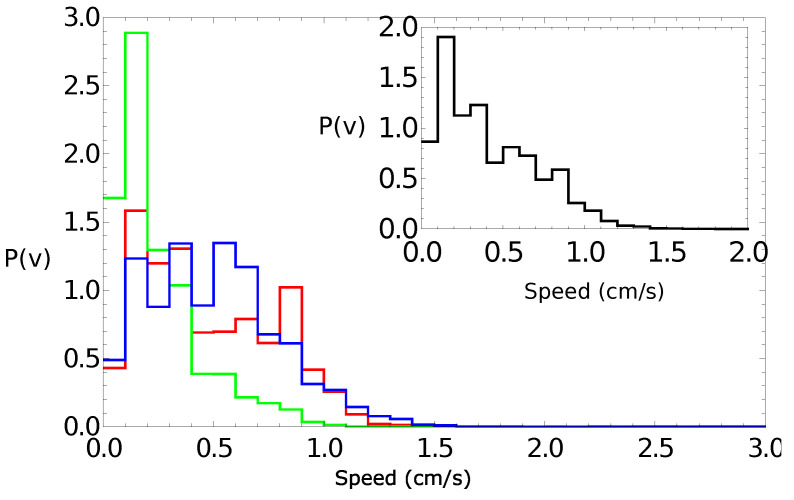
The probability distribution of the speeds for a pill made of 95% camphene and 5% polypropylene. The large figure shows the results of individual experiments. The distribution of the speed values observed in all experiments is shown in the insert.

**Figure 6 molecules-26-03116-f006:**
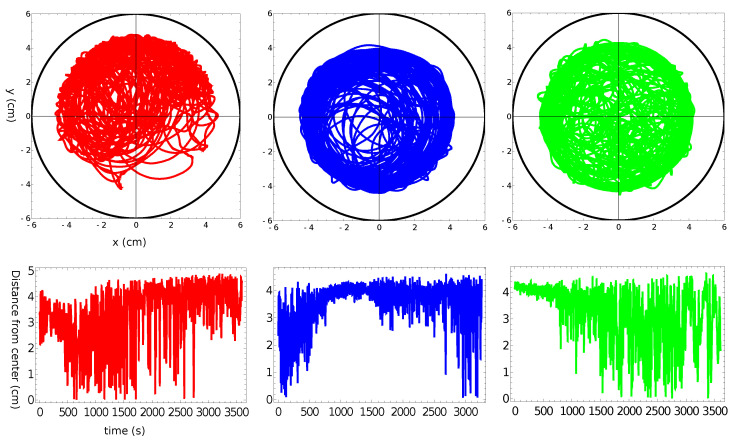
The upper three figures show the trajectories of a 4 mm pill made of 10% polypropylene and 90% camphene in a Petri dish with a 12 cm diameter. The black circle indicates the location of the dish wall. The distances between the pill center and the dish center as functions of time are plotted in the figures located below the corresponding trajectories.

**Figure 7 molecules-26-03116-f007:**
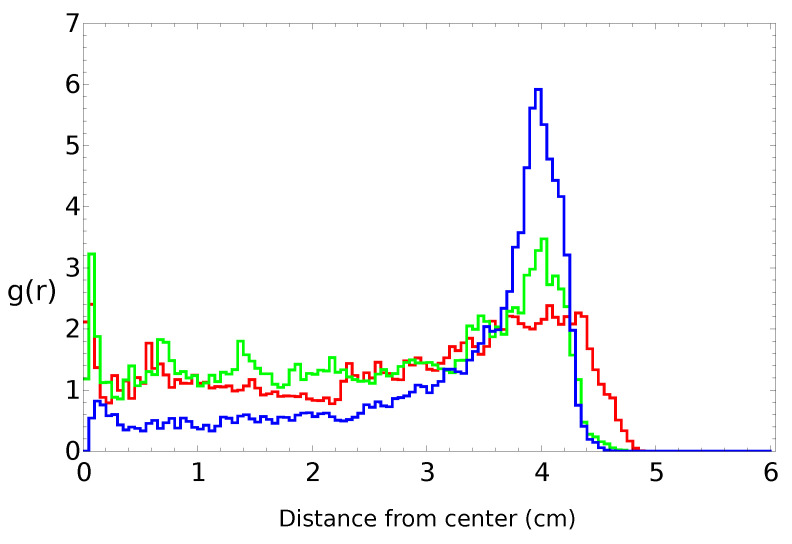
The radial distribution function of the positions of pill centers calculated from the distances illustrated in Figure 6. The same color is used to draw the radial distribution function and the trajectory.

**Figure 8 molecules-26-03116-f008:**
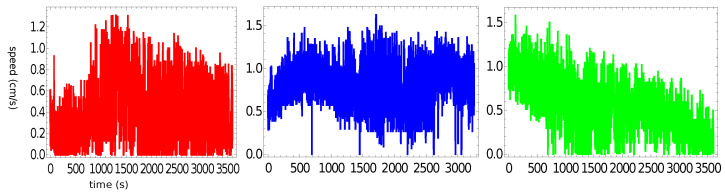
The speed of a pill made of 10% polypropylene and 90% camphene as a function of time, as observed in the three independent experiments.

**Figure 9 molecules-26-03116-f009:**
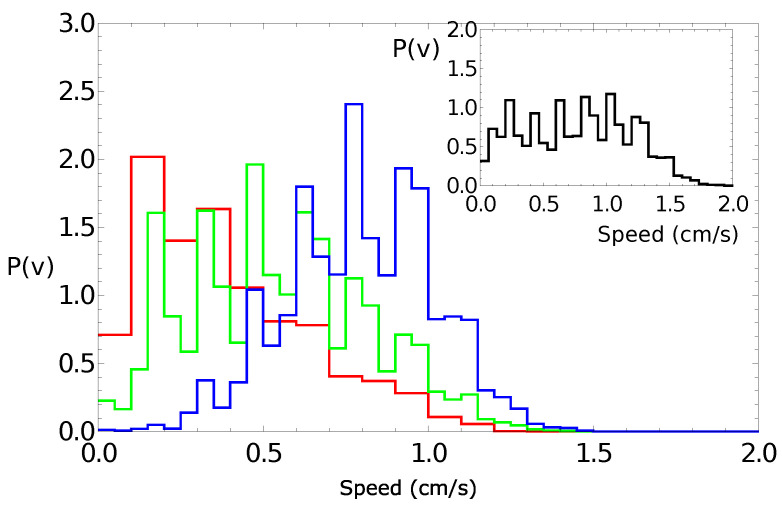
The probability distribution of the speeds of a pill made of 10% polypropylene and 90% camphene. The large figure shows the results of individual experiments. The distribution of the speed values observed in all experiments is shown in the insert.

**Figure 10 molecules-26-03116-f010:**
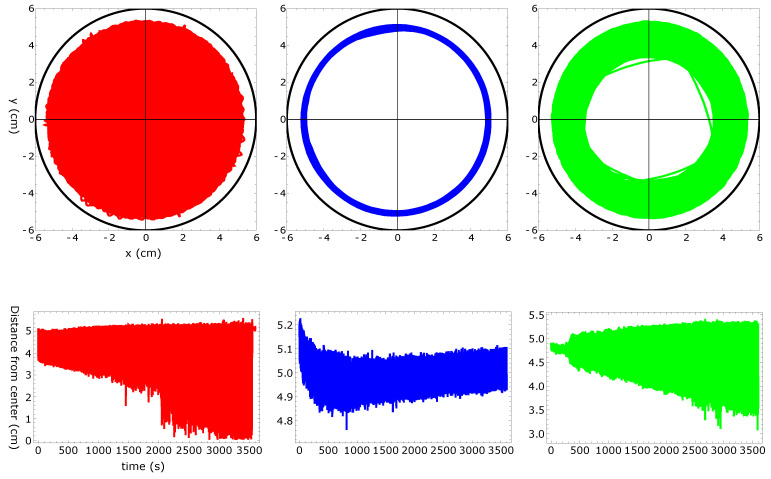
The upper three figures show the trajectories of a 4 mm pill made of 10% polypropylene, 45% camphene, and 45% camphor in a Petri dish with a 12 cm diameter. The black circle indicates the location of the dish wall. The distances between the pill center and the dish center as functions of time are plotted in the figures located below the corresponding trajectories.

**Figure 11 molecules-26-03116-f011:**
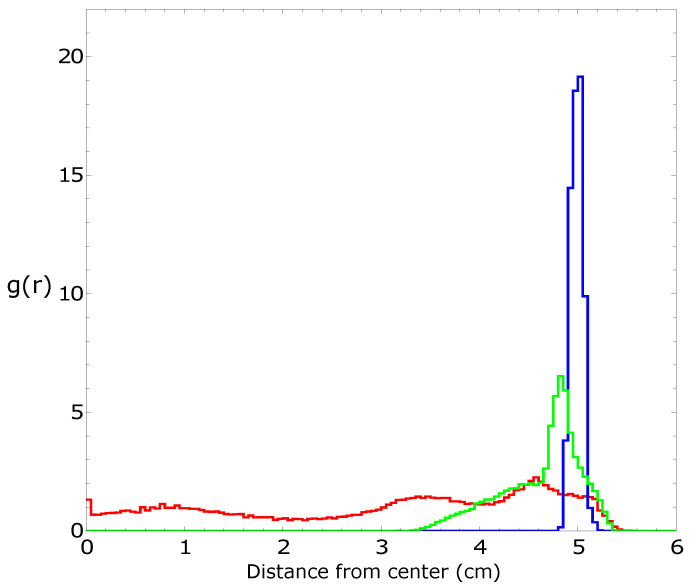
The radial distribution function of the positions of pill centers calculated from the distances illustrated in Figure 10. The same color is used to draw the radial distribution function and the trajectory.

**Figure 12 molecules-26-03116-f012:**
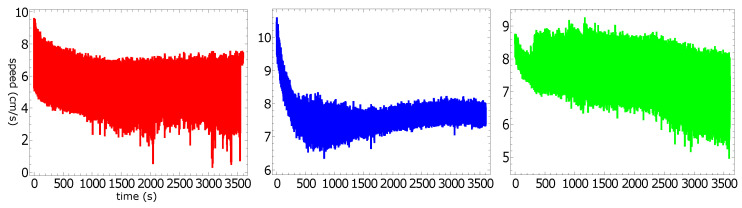
The speed of a pill made of 10% polypropylene, 45% camphene, and 45% camphor plastic as functions of time observed in the three independent experiments (cf. Figure 10).

**Figure 13 molecules-26-03116-f013:**
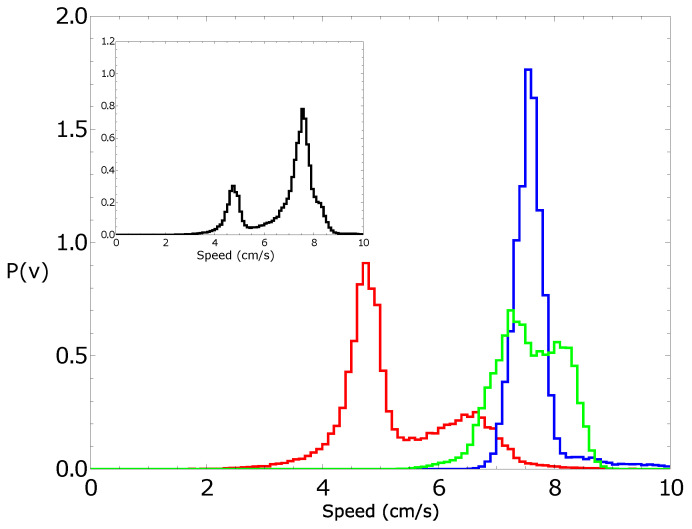
The probability distribution of the speeds. The large figure shows the results of individual experiments. The distribution of the speed values observed in all experiments is shown in the insert.

**Figure 14 molecules-26-03116-f014:**
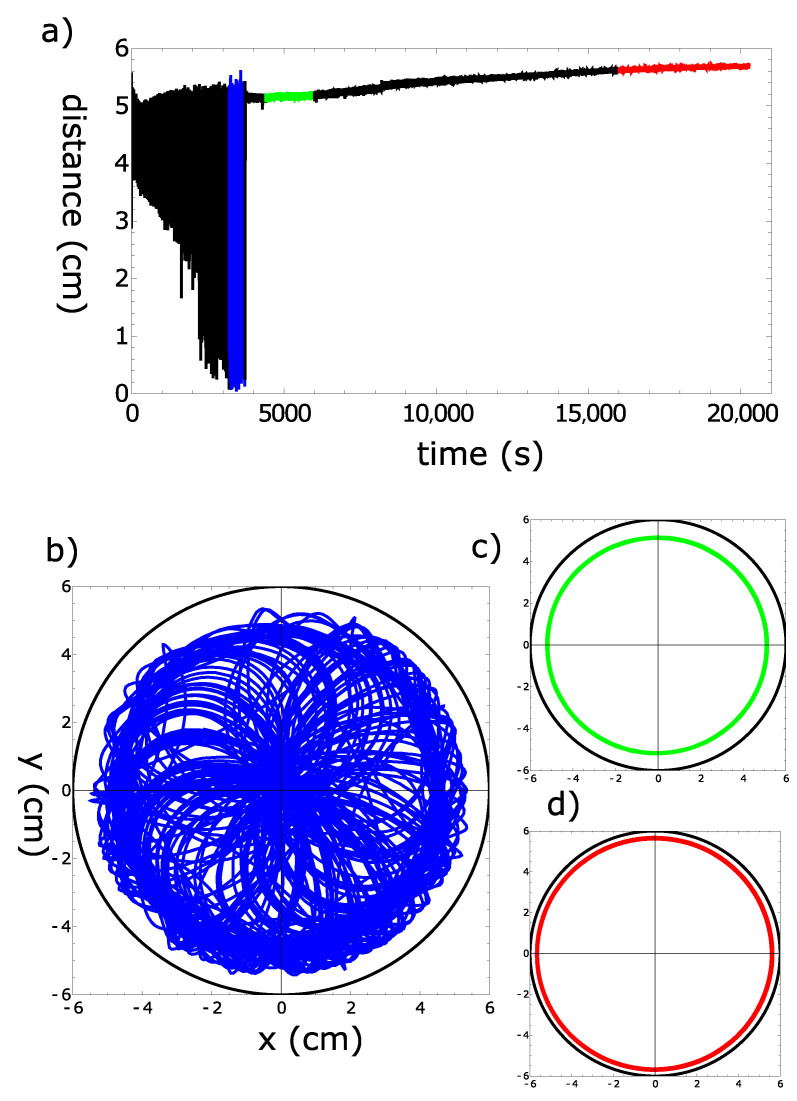
Long-time evolution of a 4 mm pill made of camphene–camphor–polypropylene plastic with a 45% camphene, 45% camphor, and 10% polypropylene weight ratio. (**a**) The distance between the pill center and the dish center as a function of time. The color fragments correspond to the trajectories shown in the figures located below. (**b**–**d**) Trajectories observed in time intervals [3200 s, 3700 s], [4400 s, 6000 s], and [16,000 s, 20,200 s], respectively. The black circles in (**b**–**d**) indicate the location of the dish wall.

**Figure 15 molecules-26-03116-f015:**
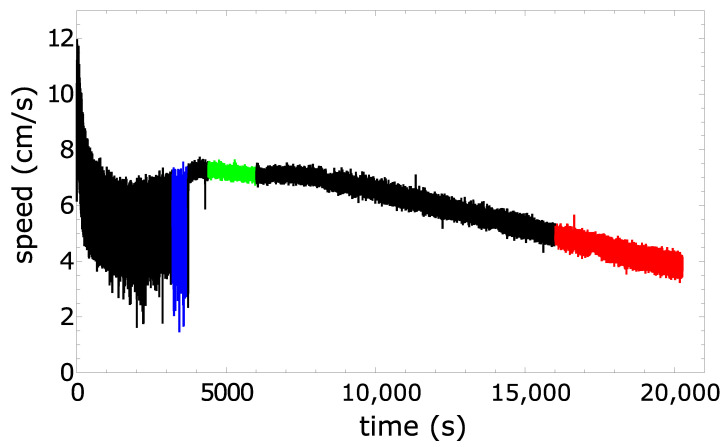
The speed of a 10% polypropylene, 45% camphene, and 45% camphor weight mixture as a function of time obtained from the trajectory illustrated in Figure 14a. The colored parts of the plot correspond to the modes of motion illustrated in Figure 14b–d.

**Figure 16 molecules-26-03116-f016:**
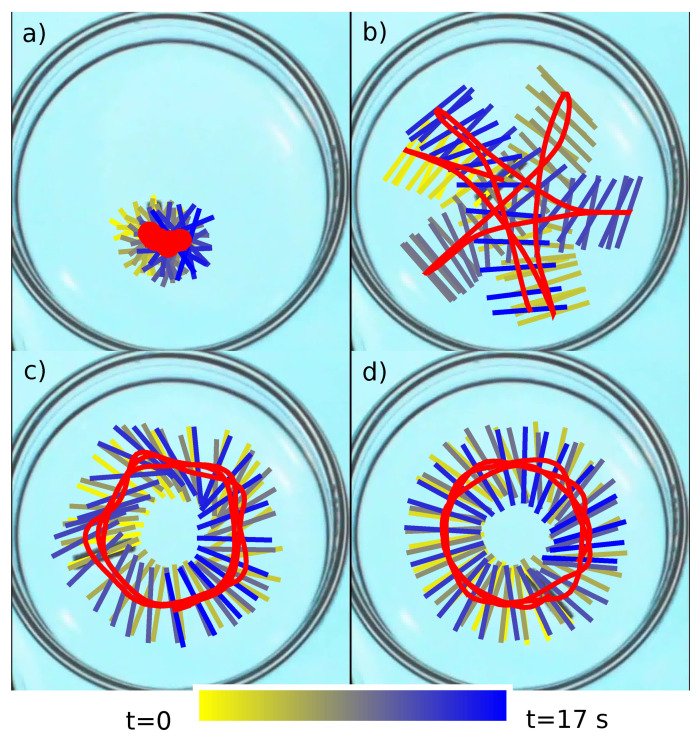
A few modes of the motion of a 1 cm-long rod made of 10% polypropylene, 45% camphene, and 45% camphor in a Petri dish with a diameter of 5 cm. The red curve marks the 17 s-long evolution of the rod center. The lines indicate the rod orientation. Time is indicated by the line color as defined by the bar below the figure. The time evolutions shown in Sub-figures (**a**–**d**) start 72 s, 240 s, 920 s, and 1200 s after the rod was placed on the water surface.

**Figure 17 molecules-26-03116-f017:**
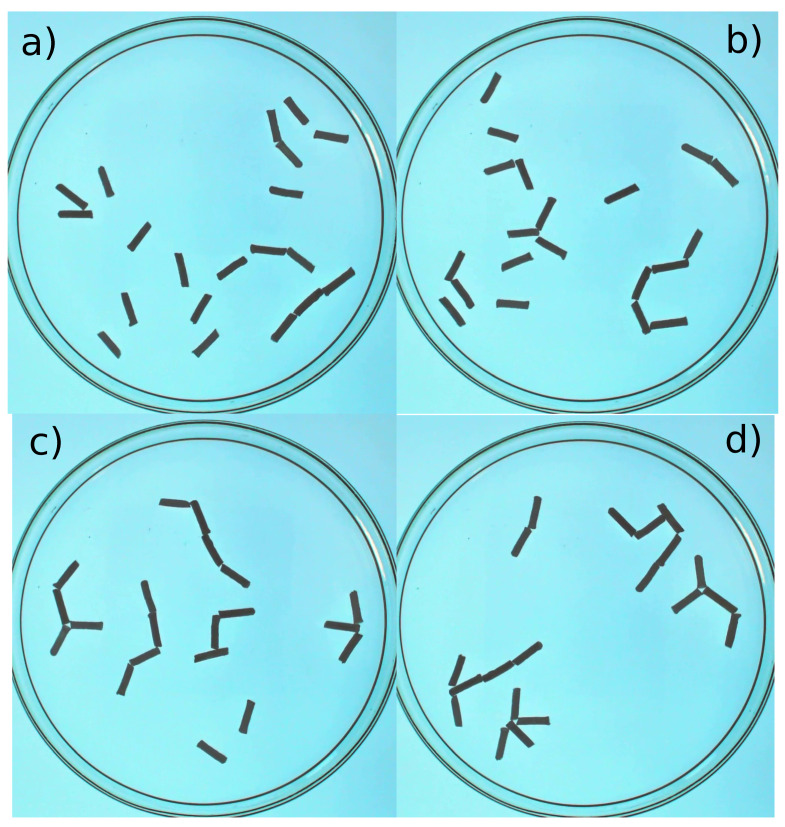
The time evolution of 20 rods that were 1 cm long on the water surface in a Petri dish with the diameter of 19 cm. The rods were made of 10% polypropylene, 45% camphene, and 45% camphor. The snapshots (**a**–**d**) correspond to the times—5 s, 30 s, 100 s, and 200 s—after all rods were placed on the surface. The movie showing the time evolution is included in the Appendix A as 20-rod-evolution.avi.

**Figure 18 molecules-26-03116-f018:**
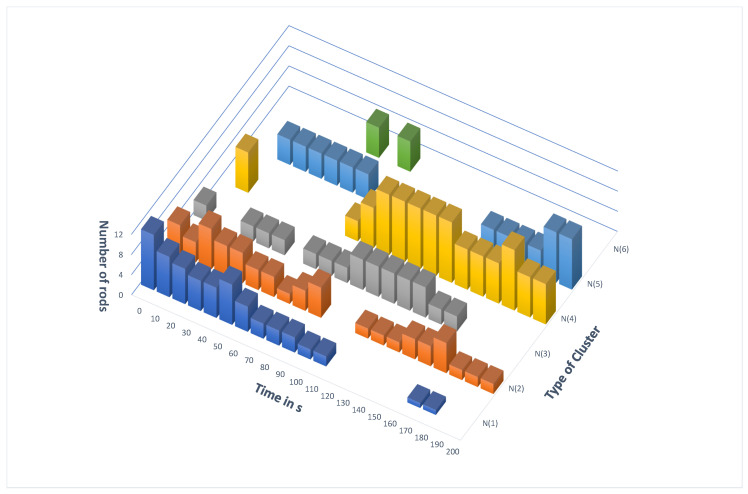
The number of rods that were linked together in clusters of s given size for times in the [0 s, 200 s] range. The results were obtained from the analysis of the movie 20-rod-evolution.avi and match Figure 17.

## Data Availability

The data are available from the Appendix A and the corresponding author.

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
