# Peer review of "A Perfect Plastic Material for Studies on Self-Propelled Motion on the Water Surface"

_molecules, 2021, doi:10.3390/molecules26113116_

Round 1

Reviewer 1 Report

The manuscript entitled “A perfect plastic material for studies on self-propelled motion on the water surface” is a continuation of the artice “A hybrid camphor-camphene wax material for studies on self-propelled motion” published as Phys Chem Chem Phys 21 (2019) 24852 by the same authors. In addition to this previous paper the current manuscript describes the preparation of a composite material of camphor, camphene and polypropylene for the sake of stability and the formation of a specific shape of the specimen. The observation of the motion (location and velocity) of the material on a water surface in petri dishes of different diameters was performed using the same approach.

The study of the phenomenon of self-propulsion of materials and the observation of a regular pattern is indeed of interest. The major part of the manunscript is well structured and presents the observed results for samples with different bulk composition very clearly. However, the questions that arise for a reader who is not very familiar with the subject of the self-propulsion of materials (and probably also for readers who are more experienced in this subject) relate to the mechanism that leads to the observed motion and an explanation for the regular average pattern that the moving samples generate. The continuous dissolution of campher molecules and the differences in surface tension between material and water as well as its temporal changes are mentioned in the manuscript. It would be beneficial to accompany the observed motion of camphor-camphene-polypropylene specimen with the analysis of changes in the surface tension of water. Furthermore, the obvious repulsion of the moving objects from the edges of the Petri dishes seeks for a hypothesis that may help to explain/understand the interesting behaviour.

In chapter 3.3 the authors mention the formation of a highly porous polypropylene foam that arises from a complete dissolution of the “volatile” compounds camphor and camphene of the composite material. This new porous material is certainly worth to mention and even to investigate in more detail. It is therefore recommended to discuss this material and a series of its properties that may be relevant for some potential applications in a separate paper. It is, however, reasonable to mention the porous polypropylene foam and the related SEM images shown in Figure 19 but rather in the supplementary information to this manuscript than as a separate chapter that is hardly linked to the main topic of this manuscript given by its title.

Since the manuscript is well written there are hardly any corrections required. Here is what I have found:

Line 93: “beaker” instead of “breaker”

Line 133: According to the subsequent equation is should read m-th frame instead of n-th frame.

Line 153: “denote” instead of “denotes”

Line 194: delete “with a”

Line 212: high-speed bursts

Line 315: “in comparison to” instead of “in compared to”

Line 406: delete one of the “that”

Figure caption 19: “Lower: The composition…” instead of “Lower at The…”

Author Response

We are grateful to Reviewer #1 for helpful comments.

Here is our answer to the comments.

  1. "It would be beneficial to accompany the observed motion of camphor-camphene-polypropylene specimen with the analysis of changes in the surface tension of water

It can be quite difficult to measure the surface tension for a freely moving object like a pill or a rod because the trajectory is unpredictable. However, direct observations of changes in the interfacial tension were reported for a rotating hexagon powered by 6 camphor disks located at its corners (Nakata et.al, doi:10.1021/acs.jpcc.7b12089), for which the trajectory was well defined. It has been measured that the water surface tension very close to a pill was ~ 65 mN/m, whereas far away from a pill, it is around 70 mN/m. This information has been added to the Introduction, lines 31-36:

There is a continuous flux of camphor molecules that modifies the surface tension from the source to the air through the surface layer. The surface tension remains low in the region close to the camphor piece, where the concentration of camphor is high.  The surface tension has been estimated as 65 mN/m proximal to the source and 70 mN/m distal from the camphor \cite{Suematsu2014,kara,Nakata2018}.

  1. "Furthermore, the obvious repulsion of the moving objects from the edges of the Petri dishes seeks for a hypothesis that may help to explain/understand the interesting behaviour."

The stable motion along the edge can be explained as follows: We can assume that the surface active molecules are symmetrically dissipated around the disk. Those that move towards the dish center are dispersed over a larger area than those that move towards the dish edge because the edge restricts their motion. As a result, one can expect a higher concentration of surface active molecules in the region between the pill and the dish edge than in the central part of the dish. The gradient of concentration translates into the gradient surface tension that repels a pill away from the edge.

The comment given above has been added to the manuscript in section 3.1, lines 209-215.

  1. "In chapter 3.3 the authors mention the formation of a highly porous polypropylene foam that arises from a complete dissolution of the “volatile” compounds camphor and camphene

of the composite material. This new porous material is certainly worth to mention and even to investigate in more detail. It is therefore recommended to discuss this material and a series of

its properties that may be relevant for some potential applications in a separate paper. "

We agree with this suggestion. We have removed Section 3.3 from the manuscript. We will prepare another publication with information on polypropylene foams, their structures, and properties.

All suggested language corrections have been introduced.

Reviewer 2 Report

The study is about plastic materials containing camphor, camphene and polypropylene which have a self-propelled motion. I do not think that this research has a novelty to be published in Molecules.  For improvement of the research paper, I made some comments.

  1. In the abstract, please comply reference guide of the journal.
  2. To support the author's statement in the introduction, references should be added.  
  3. All of the graphs should be improved.
  4. In Figure 2, 3, 4,5, 6, 7, 8, 9, 10,11, 12, 13, It seems that the author did three repeating tests, but each graph is from one sample test in the figures. Is there any special reason? If it were not, please combined them into one graph and make an average.
  5. Statistical analysis is required for experimental results.
  6. Please correct mistyping and abbreviation of journal name in references
  7. The sample including camphene, polypropylene and camphor showed a distinctive motion compared to others. But the author needs to statistically compare it with the others.
  8. The manuscript needs a professional proofreading.

Author Response

The reviewer wrote:  " I do not think that this research has a novelty to be published in Molecules".

    I do not think the reviewer is right. As far as I know, the hybrid material composed of camphene, camphor, and polypropylene has not been studied in the context of self-propelled motion. Could the reviewer point out any other publication in which a similar material was described? 

No other reviewer questioned the novelty of our manuscript. The submission of our manuscript was on the invitation of dr. Daniela Meroni: I can quote her letter:

"As Guest Editor of the Special Issue of Molecules (IF 3.267) “Hybrid Materials for Advanced Applications”, it is my pleasure to invite you to submit an article on this topic..."

  1. In the abstract, please comply reference guide of the journal.

We have modified the abstract strictly according to the Molecules Journal guidance and now it reads:

We describe a novel plastic material composed of camphene, camphor, and polypropylene that seems perfectly suited for studies on self-propelled objects on the water surface. Self-motion is one of the attributes of life, and chemically propelled objects show numerous similarities with animated motion. One important question is the relationship between the object shape and its motility.  In our previous paper, {R. Löffler et al. PCCP, 2019, 21, 24852–24856}, we presented a novel hybrid material, obtained from the solution of camphor in camphene, that allows making objects of various shapes. This hybrid material has wax-like mechanical properties, but it has a very high tackiness. Here we report that a small amount of polypropylene removes this undesirable feature. We investigate the properties of camphor-camphene-polypropylene plastic by performing the statistical analysis of a pill trajectory inside a Petri dish and compare them with those of camphor-camphene wax. The plastic shows the stable character of motion for over an hour-long experiment. The surface activity of objects made of plastic does not significantly depend on the weight ratios of the compounds. Such a significant increase in usefulness comes from the polypropylene that controls the dissipation of camphor and camphene molecules.

  1. To support the author's statement in the introduction, references should be added.

A number of references have been added to the introduction.

  1. All of the graphs should be improved.

Mr./Ms. Reviewer, could you please be more specific with your request?

  1. In Figure 2, 3, 4,5, 6, 7, 8, 9, 10,11, 12, 13, It seems that the author did three repeating tests, but each graph is from one sample test in the figures. Is there any special reason? If it were not, please combined them into one graph and make an average.

In the manuscript, we have clearly written that subfigures marked by different colors correspond to different experiments with pills made of the same material (see p.6). The presentation of individual results is supposed to demonstrate the different characteristics of motion observed in different experiments, related to the existence of different multistable modes of motion, as discussed in the manuscript.  The combined plot would not allow for a clear presentation.

  1. Statistical analysis is required for experimental results.

The statistical analysis is given in Figures 3,5,7,9,11, and 13. The quantitative measures of the analysis (the maximum approach to dish edge, average speed) discussed in the text.

  1. Please correct mistyping and abbreviation of journal name in

references

We have corrected a number of mistakes.

  1. The sample including camphene, polypropylene and camphor showed a distinctive motion compared to others. But the author needs to statistically compare it with the others.

We have done such analysis using the radial distribution functions for pill positions with respect to the dish center and the probability distributions of speeds (Figures 3,5,7,9,11 and 13). The comparison is presented and discussed in the manuscript.

  1. The manuscript needs a professional proofreading.

We have introduced language corrections suggested by the other referees. Moreover, the English of the text has been proofread by a native speaker.

Jerzy Górecki

Reviewer 3 Report

Page 1 Line 30. Extra space is presented in the parenthesis

Page 3, Line 119, the section “pill moving on the water surface” should be described in the methodology section, not only as results and discussion.

Page 3, Line 132, The same comment as the last one, the mathematical study and analysis must be provided in the methodology section.

Page 5 In Figure 1, not only a graphical type of pill displacement has to be presented. Please add a picture of the pills before and after the movement.

Page 6 Figure 2, please describe how the camera images were possible to build the proper figures. In lines 197-205 is briefly described. Please be more specific.

Finally, some chemical characterization of the pills must be provided, such as IR analysis and morphological images such as SEM or TEM. And its relationship with the presented results. 

Author Response

We are grateful to Reviewer #3 for the report.

Here is our answer to the comments.

  1. Page 1 Line 30. Extra space is presented in the parenthesis

The extra space has been deleted.

  1. Page 3, Line 119, the section “pill moving on the water surface” should be described in the methodology section, not only as results and discussion.

The section has been shifted to the methodology section.

  1. Page 3, Line 132, The same comment as the last one, the mathematical study and analysis must be provided in the methodology section.

The section has been shifted to the methodology section.

  1. Page 5 In Figure 1, not only a graphical type of pill displacement has to be presented. Please add a picture of the pills before and after the movement.

We have re-drawn the figure. The pill locations at both ends of the trajectory have been added.

  1. Page 6 Figure 2, please describe how the camera images were possible to build the proper figures. In lines 197-205 is briefly described. Please be more specific.

We have added the following explanation to the methodology section;

The movies with recorded experiments were separated into frames using the ffmpeg \cite{web} program. The locations and orientations of objects on individual frames were extracted using ImageJ \cite{ImageJ} or procedures of Mathematica \cite{Math}. The information on the object time evolution (trajectory and speed as functions of time) was analyzed with Mathematica.

  1. Finally, some chemical characterization of the pills must be provided, such as IR analysis and morphological images such as SEM or TEM. And its relationship with the presented results

This comment is essential for the future, more detailed study. There are no reactions expected between camphene, camphor, and polypropylene. We also do not expect significant differences in dissipation between camphene and camphor because the molecules are similar. Our opinion is supported by the stability of observed speed as a function of time. Nevertheless, it would be interesting to analyze the pill composition as a function of time, and we plan to do such experiments in the future. Still, not within 10 days, we are given to prepare the corrected version of the manuscript. Following the suggestions of Reviewer #1 that the manuscript should be focused on self-propelled properties, we decided to delete Sub-section 3.3 and prepare another paper concerned with the properties of foams obtained for different compositions of plastics